# Antibiotic Prophylaxis in One-Stage Revision of Septic Total Knee Arthroplasty: A Scoping Review

**DOI:** 10.3390/antibiotics12030606

**Published:** 2023-03-18

**Authors:** Carlo Ciccullo, Thomas Neri, Luca Farinelli, Antonio Gigante, Rémi Philippot, Frederic Farizon, Bertrand Boyer

**Affiliations:** 1Clinical Ortopaedics, Department of Clinical and Molecular Sciences, Università Politecnica delle Marche, 60121 Ancona, Italy; 2Department of Orthopaedic Surgery, University Hospital of Saint Etienne, Hôpital Nord, 42055 Saint-Étienne, France; 3Laboratoire Interuniversitaire de Biologie de la Motricité, Université de Lyon, 69361 Lyon, France; 4U 1059 Sainbiose, Mines Saint-Étienne, Universitè Jean Monnet, INSERM, 42023 Saint-Étienne, France

**Keywords:** periprosthetic joint infection, TKA, antibiotic prophylaxis, revision TKA, one-stage surgery

## Abstract

Background: Total knee replacement (TKA) is becoming a routine procedure in orthopedic surgery. One of the possible complications of this surgery is periprosthetic joint infection (PJI). The purpose of this study is to identify, through a literature review, which antibiotic is used as prophylaxis for septic one-stage revision TKA and what is the rationale for its use. Methods: We searched: MEDLINE, Embase, PsycINFO on Ovid, the Cochrane Library, and the Google Scholar Database. The searches were limited by date (January 2005 to September 2022) and to the English language. All types of original research were considered, including prospective or retrospective longitudinal studies, cross-sectional studies, and randomized trials. The specific search terms were ((antibiotic [MeSH]) AND (prophylaxis)) and (TKA OR TKR OR “Arthroplasty, Replacement, Knee” [MeSH] OR ((knee) adj2 (replace* OR arthroplasty* OR prosthe*))). Results: Despite our research efforts, we found no article capable of answering the question of which antibiotic to use as surgical prophylaxis for a septic revision one-stage TKA. Conclusions: Although the research results are inconclusive, we would recommend using the same antibiotic prophylaxis as for primary joint replacement, i.e., cefazolin, as it was recommended for its low side effect rate and relative effectiveness.

## 1. Introduction

Total knee arthroplasty (TKA) has become the option of choice both in terms of cost and success in the treatment of knee osteoarthritis (OA) [1,2], one of the most terrible complications is periprosthetic joint infection (PJI). This is one of the major causes of revision surgery. Kurtz et al. found that the relative incidence of PJI ranged between 2.00% and 2.40% of total knee arthroplasties (TKA) [3]. According to Springer et al., about 1.03% of TKAs were revised due to infection [4,5,6]. PJI was associated with a substantial financial burden on the healthcare system and a significant physical and psychological morbidity in patients [3,4,5,6,7,8,9].

In 2011, the Musculoskeletal Infection Society (MSIS) and the International Society for Infectious Diseases (ISID) developed criteria to standardize the definition of PJI [8,9]. In 2018, Parvizi et al. [5] introduced new diagnostic criteria for the identification of PJI to address the limitations of prior definitions. They divided the criteria into two categories: Major and Minor. The Major criteria are: the identification of the same pathogen in at least 2 samples and the presence of sinus that communicated with the prosthesis. The Minor criteria are: a single positive culture, elevated serum C-reactive protein (CRP) and erythrocyte sedimentation rate (ESR), purulence in the affected joint, positive histological analysis of periprosthetic tissues, elevated synovial fluid WBC count and elevated synovial fluid PMC%.

Nowadays, many studies are still in progress to try to expand our knowledge on the diagnostics of this issue (the current search for synovial biological markers is one diagnostic method with the most possible developments for the future) [6,10].

The management of PJI is complex and reflects the multifactorial nature of the problem. Unfortunately, antibiotic therapy alone does not lead to satisfactory results or the complete recovery of the patient. Therefore, revision surgery through one or two stages might be performed. However, in the elderly, PJI surgery may result in a higher incidence of mortality [7,11,12].

One-stage revision for septic prosthesis has been studied and reported in a few centers [13,14,15,16,17,18,19]. Arguments about the advantages of this procedure focus on the fact that only one surgery is needed, which lowers costs, provides a more rapid restoration of function, and possibly lowers the rate of morbidity. Knowing the species of microorganisms before performing a one-stage revision is recommended; thus, specialized teams often decide on the antibiotic regimen before the intraoperative sampling. However, revision surgery still shares an intrinsic risk of infection; therefore, antibiotic prophylaxis is still recommended, which may result in a possible conflict between antibiotic prophylaxis and antibiotic therapy.

Prophylactic antibiotics reduce the risk of developing PJI [20,21,22,23,24]. Nowadays, cephalosporins are the primary choice for antibiotic prophylaxis in first-intention TKA. However, with the increasing emergence of bacterial resistance, the need to use a new antibiotic class as prophylaxis, i.e., vancomycin, has increased. In recent years, to reduce the rate of postoperative infections, there have been attempts to use new antibiotic classes, which have, however, proved to be as effective as the current cefazolin prophylaxis [25,26,27,28,29,30].

Looking at our experience in the revision of septic TKAs, we looked at the best management of patients undergoing one-stage surgery at the level of antibiotic prophylaxis and how this antibiotic management could act at the level of clinical outcome. The purpose of this article is to find, through a scoping review of the literature, which class or type of antibiotic used as prophylaxis for one-stage septic revision TKA might be recommended to lower the risk of reinfection, and what rationale, if any, or body of evidence supports the use of specific antibiotics for the prophylaxis of surgery in one-stage revision.

## 2. Results

The initial search identified 746 studies. After removing duplicates and studies not meeting the inclusion criteria, the review of titles and abstracts yielded 74 potential studies. All the selected studies were excluded during the full-text review for not describing antibiotic selection and administration, not reporting on PJI, or not including a control group (see PRISMA flow diagram [31] in Figure 1 for included and excluded studies).

Despite a thorough search through all available studies on single-stage surgery, we were not able to find any evidence with regard to the use of antibiotic prophylaxis in infected TKA one-stage revision.

During our research, we found that many authors focused on the use of antibiotic prophylaxis in aseptic prosthesis replacement.

## 3. Discussion

The administration of preoperative prophylactic antibiotics represents the standard of care for primary TKAs, having been shown to significantly reduce the risk of periprosthetic joint infection [32,33,34,35].

However, some studies reported that the use of antibiotic prophylaxis for suspected septic TKAs might alter culture results, making a diagnosis and targeted antibiotic therapy more difficult [36,37].

Tetreault Ba M. et al. [38] and Stephen et al. [39] demonstrated in their studies how the administration of antibiotic prophylaxis does not alter the intraoperative tissue culture in both TKA and hip septic and aseptic revisions.

Ghanem et al. [40] analyzed 171 knees, 72 of which received antibiotic prophylaxis and 99 did not. The study showed how the intraoperative cultures carried a comparable number of false negatives (12.50% of the 72 and 8.00% of the 99).

The microbiological and resistance epidemiology of periprosthetic joint infections varies between countries. In the USA, the most common organisms are methicillin-resistant and methicillin-sensitive *S. aureus*, and methicillin-resistant and methicillin-sensitive *S. epidermidis* [41,42]. Europe showed the highest prevalence of coagulase-negative *Staphylococcus* spp., followed by *S. aureus*, streptococcus, and enterococcus organisms [43,44]. Asia showed the highest prevalence of aerobic Gram-positive bacteria, which includes *S. aureus*, followed by coagulase-negative *Staphylococcus* and Gram-negative bacteria [45].

In the study by Roman et al., it was stated that all cases of acute/subacute haematogenic PJI were caused by aerobic and microaerophilic Gram-positive pathogens. Both *Staphylococcus epidermidis* and *methicillin-resistant S. Aureus* were found in 91.66% of haematogenic PJI cases. In this context, empirical antimicrobial therapy for acute PJI should focus on aerobic or microaerophilic Gram-positive cocci [46].

Nickinson et al., in their study, described the occurrence of coagulase-negative staphylococci dominating the group examined (49%) [47,48].

Drago et al. evaluated the microbiological findings of approximately 429 knee and hip PJIs, showing that staphylococci were the most frequent organism in 66.6% of cases, followed by *Enterobacteriaceae* and *Cutibacterium acnes* [48,49].

Antibiotic prophylaxis is used to prevent bacterial overinfection [50]. From a pharmacological point of view, antibiotic prophylaxis aims to maintain drug levels in serum, tissue, and bone at the lowest inhibitory concentration throughout surgery to avoid the attachment of possible microorganisms [30]. The most common antibiotics used in orthopedic surgery are bactericidal: penicillin, cephalosporins, and vancomycin [51]. Cephalosporins are the most common prophylactic antibiotics in primary arthroplasty [29].

Antibiotic prophylaxis (AP) should prevent surgical site contamination from developing into an infection. Effective antibiotic prophylaxis, however, must have the proper timing, dosage, and choice of antibiotic [52,53].

Recent studies suggest that the optimum timing for the preoperative antibiotic dose should be within less than 1 h of the skin incision and extended postoperatively for a duration of 24 h [54].

Parsons et al. [55] studied forty-two international orthopedic societies to try to find out which antibiotic prophylaxis was the most widely used. The study found that only fourteen societies had published guidelines or statements regarding presurgical antibiotic prophylaxis through websites. Twenty-five companies had no visible guidelines on their website. The publication date of the guidelines ranged from 2011 to 2020. 59% of the studied guidelines suggested a first-generation cephalosporin, such as cefazolin, as a first line. A total of 5% of guidelines recommended cloxacillin, and 35% did not specify agents, but suggested broad-spectrum intravenous antibiotics. Five of the ten guidelines proposing second-line agents suggested vancomycin in cases of known allergy, while the other five proposed clindamycin as a second-line treatment. The Australian Orthopedic Association (AOA) suggests teicoplanin as another alternative. The Australian, Philippine, and Thai arthroplasty societies recommend vancomycin as the agent of choice in cases of “high risk of MRSA”. The British “Getting It Right First Time” (GRIFT) initiative and the American Association of Hip and Knee Surgeons (AAHKS) recommend the use of “broad spectrum antibiotics” without specifying agents.

Wyles et al., in their study, analyzed a large group of patients (29,695 patients) and reported that PJI rates were significantly higher when noncefazolin antibiotics were used for the preoperative prophylaxis of TKA and THA [52].

Most antibiotic prophylaxis guidelines encourage a “single dose policy”, although the prolongation of AP to 24 h and beyond is still a clinical method in some countries. Both registry data and observational clinical studies do not support the superiority of multiple doses in routine procedures, except for long procedures and those requiring major blood transfusions [56].

Most published guidelines suggest the use of intravenous broad-spectrum antibiotics. The most common organisms that need more attention, according to the guidelines, are *Staphylococcus aureus, Staphylococcus epidermidis, Escherichia coli*, and *Proteus* [57].

First- and second-generation cephalosporins lend themselves well to prophylaxis, having excellent coverage of Gram-positives and Gram-negatives. Third generation cephalosporins are not recommended in any guidelines regarding total joint arthroplasty, as they have reduced activity on Gram-positives. Cloxacillin is recommended less, despite its low side effect profile, due to its narrow spectrum of activity against the more common organisms associated with PJIs [58,59].

According to the American Academy of Orthopedic Surgeons (AAOS), the routine use of vancomycin could promote the development of vancomycin-resistant enterococci (VRE) colonizations and infections. It has also been recommended that vancomycin be reserved for the treatment of serious infections with beta-lactam-resistant organisms or for the treatment of infections in patients with a life-threatening beta-lactam antimicrobial allergy [60].

In an attempt to reduce the rate of TKA infections, several studies have compared the efficacy of single- versus dual-antibiotic prophylaxis (e.g., cefazolin and vancomycin). All the studies we analyzed on this topic were retrospective and the results remain inconclusive; they did not demonstrate any superiority of dual- over single-antibiotic prophylaxis [31,54].

The study by Sewick et al. [61] showed an equal incidence of infections (*p* = 0.636) in patients undergoing primary TKA who had received cefazolin alone and those who had received antibiotic prophylaxis with cefazolin and vancomycin.

To our knowledge, this is the first study to evaluate the choice of antibiotic class as prophylaxis for one-stage septic total knee arthroplasty (TKA) revision surgery and its potential beneficial effect in reducing cases of postoperative reinfection.

It was interesting to note that although there were no developments in the literature on the topic we researched, many authors studied what was the best antibiotic prophylaxis for aseptic revisions of TKA.

The surgical revision of TKA can be performed in one or two stages. A two-stage revision of septic implants is the most common procedure for the treatment of infected prostheses [62,63,64,65,66,67]. It was described for the first time by Insal et al. in 1983 [68]. The treatment involves the removal of the prosthesis, followed by extensivedebridement of nonviable soft tissues, a synovectomy, copious irrigation and lavage, and the reaming of the medullary canals. Once the joint is prepared, cement beads and/or an antibiotic-loaded cement spacer are applied [69]. Postoperatively, antibiotics are administered according to the sensitivity of the infecting organisms. Prosthetic reimplantation is performed after the end of the antibiotic course, once the wound has healed and a successful antibiotic treatment can be confirmed. During the second stage, the removal of the microspheres or spacer is performed, and following copious irrigation and washing, further debridement and implantation of the new prosthesis is performed [70]. According to several authors, the success rate of this type of surgery for a knee replacement infection is between 91.00% and 96.00% [16,17,18,19,20,21,22,63,64,65,66,67,68,69,70,71].

One-stage surgery is less frequently performed in the United States than two-stage arthroplasty. In this procedure, a major arthrotomy with debridement is performed and then the prosthesis is removed. The debridement is very radical and aggressive and involves not only soft tissue but also portions of the bones that are in contact with the prosthetic element. Next, the surgical site is washed thoroughly with water and antimicrobial solution through a pulsed wash. Then, the new prosthesis is placed. In 1991, Von Foerster et al. [71] reported an infection control rate of 73.00% in 104 patients treated by one-stage replacement surgery. According to Romano et al., success rates at about 40 months after surgery are 81.90%, while according to Klouche et al., success rates are 100.00% at more than 2 years of follow-up [72,73]. Other retrospective studies have been performed over the years and have shown an uptick in success rates of up to 95.00% [14,15,17,18,74,75], which is comparable to the rates resulting from two-stage revision.

Several reviews have been performed in recent years to assimilate the evidence on single-stage revision TKA for infection. Niagra et al. [76] found no differences in reinfection rates between one- and two-stage procedures in their systematic review of five cohort studies. Kunutsor et al. [77] compared ten single-stage studies with 108 two-stage studies for generally unselected patients and found similar reinfection rates of 7.6% and 8.8%, respectively, in their meta-analysis. Pangaud et al. [78] performed a systematic review of 14 single-stage articles involving 687 patients and 18 two-stage articles involving 1086 patients. The mean reinfection rate was 12.90% for single-stage revision and 15.20% for two-stage revision, with a similar function across both groups.

Most success attributed to one-stage surgery is due to a total synovectomy, the careful debridement of the surrounding soft tissues, and the administration of therapy [79].

The indications and contraindications of this type of surgery, listed by the International Consensus Meeting on Periprosthetic Joint Infection, are: (1) Indications: A non-immunocompromised host, an absence of systemic sepsis, minimal bone/soft tissue loss allowing primary wound closure, and a known pathologic organism with sensitivities preoperatively. (2) Contraindications: A severe soft tissue defect or unresectable sinus tract, a culture-negative PJI, an inability to perform a radical debridement, an inability to deliver local antibiotic treatment, and a lack of bone stock for the fixation of the new implant.

In his study, Fracs et al. [80] analyzed the amount of vancomycin absorbed into the bone by 10 patients undergoing aseptic revision of a TKA with prior treatment by intraosseous regional administration (IORA). In the study by Young et al. [81], they confirmed the absorption of vancomycin was greater in the group of patients treated via IORA than in those treated IV. Liu et al. [81] studied the effect of combining cefazolin and vancomycin in patients undergoing an aseptic revision of TKA to reduce the rate of PJI. Kuo et al. [82] focused their study on the 24-hour extension of antibiotic prophylaxis performed with first-generation cephalosporin and demonstrated that extended postoperative prophylactic antibiotics did not reduce the PJI rate in aseptic revision TKA compared with the standard group.

The use of extended oral antibiotics (EOA) was addressed by several authors, including Villa et al. [83]. In their study, the authors demonstrated how there were no substantial differences between the use of traditional and extended prophylaxis in the rate of PJI in the two groups analyzed. The study carried out by Zingg et al. [84] focused on 176 aseptic surgical revisions of TKAs performed between 2013 and 2017. In the study, all patients were given intravenous antibiotics during hospitalization and were discharged with 7-day oral antibiotic prophylaxis. The study demonstrated how extended oral antibiotic prophylaxis (EOA) after aseptic revision of TKA had the same PJI rates as the selective one.

Despite countless technological advances in diagnostics, even today, there is not a single definitive diagnostic test for PJI, only a work-up procedure that includes clinical, cytological, histopathological, and microbiological investigations [5,85]. Currently, the gold standard for the diagnosis of PJI is pre- and intraoperative microbiological cultures [8,86].

Synovial fluid culture has rather limited sensitivity, so it cannot rule out PJI with certainty. The reasons for low sensitivity include a low bacterial load in chronic low-grade infections, the presence of pathogens adhering to the surface of the implant and forming a biofilm, prior antimicrobial treatment, a delayed transport, or an inadequate transport of specimens [87,88]. A positive culture, on the other hand, can be determined by contamination and often does not automatically define an infection, except in cases of high-virulence pathogens that are rarely contaminating.

Pathogen identification is critical in the management of PJI as such information can guide not only the administration of perioperative antibiotics, but also treatment protocols, and can also guide the clinician in deciding a prognosis [89]. Although preoperative aspiration cultures are more convenient and easier to obtain than intraoperative cultures, this method raises the possibility of a suboptimal treatment protocol in case of discordant cultures [86,90,91,92].

Before synovial fluid aspiration, it is important to remember that patients should stop antibiotic therapy. According to some articles, antibiotic treatment should be stopped about 4 weeks before aspiration or any intervention looking for foreign agents, to avoid false-negative results [93,94,95,96].

Few studies evaluated the concordance between preoperative and intraoperative culture results, which explained these results of 63.00% to 77.00%, respectively [43,97,98,99,100].

Somme et al. [101] reported complete agreement in 63.60% of cases; in 15.90% cases, the aspirate did not recover all the organisms found intraoperatively. Buchholz et al. [102] found that agreement with intraoperative samples was 73.00% in a study that included 205 preoperative joint aspirates in cases of suspected periprosthetic infection.

According to le Vavasseur et al. [103], empiric antibiotics treatment, which usually combines a beta-lactam with activity against Gram-negatives and a drug effective against Gram-positives (including methicillin-resistant Gram-positives), could be used to tailor the treatment from empirical coverage to pathogen-specific treatment at the beginning of treatment, pending culture test results and once microbiological results are available.

This study was not without limitations. First, meta-analyses could not be performed for a number of end points because there was a lack of studies examining the particular end point. Second, some applicable studies may not have been identified by the search methodology; however, our search was comprehensive based on our use of extensive medical subject headings.

## 4. Materials and Methods

This scoping review was performed according to the Preferred Reporting Items for Systematic reviews and Meta-Analyses extension for Scoping Reviews (PRISMA-ScR) guidelines [104].

### 4.1. Inclusion and Exclusion Criteria

The study inclusion criteria consisted of studies that went on to report any form of follow-up and surveillance, either face-to-face or using questionnaires or virtual methods, regarding adults who had undergone one-stage revision surgery of infected TKAs and for whom there was indicated antibiotic therapy.

Studies that reported clinical case reports without a validated outcome, or in which surgery was not described, were excluded to avoid bias. Studies in which prophylaxis was not mentioned or in which prophylaxis was not described were also excluded. We also excluded all studies that involved the review of nonseptic TKAs.

### 4.2. Literature Research

We searched: MEDLINE, Embase, PsycINFO on Ovid, the Cochrane Library, and the Google Scholar Database. The searches were limited by date (January 2005 to September 2022) and to the English language. All types of original research study were considered, including prospective or retrospective longitudinal studies, cross-sectional studies, and randomized trials. The specific search terms were ((antibiotic [MeSH]) AND (prophylaxis)) and (TKA OR TKR OR “Arthroplasty, Replacement, Knee” [MeSH] OR ((knee) adj2 (replace* OR arthroplast* OR prosthe*))).

### 4.3. Study Selection

Two independent reviewers (C.C. and B.B.) screened all the identified titles and abstracts. All randomized controlled trials (RCTs) as well as prospective cohort, retrospective cohort, and case-control studies pertaining to one-stage total knee revision arthroplasty, antibiotic prophylaxis for total revision knee arthroplasty, and antibiotic prophylaxis for total joint revision replacement were included.

Potentially eligible studies were identified, and the full-text articles of these studies were retrieved for review. The reference lists of the full text articles and related citations were searched for relevant articles that were not identified in the original search. Our research strategy identified 74 studies (43 on Embase, 12 on Medline, and 19 on Cochrane).

## 5. Conclusions

PJI is one of the most devastating complications of TKAs. The solution to this complication is almost always based on surgery. Based on the available data, it is not possible to make recommendations on antibiotic prophylaxis regimens for septic one-stage revision TKA. Our results demonstrate the need for further level I studies with adequate power to evaluate which class or type of antibiotic used as prophylaxis for septic TKA single-stage revision could be recommended to reduce the risk of reinfection. At present, we would suggest using the same antibiotic prophylaxis as for primary joint replacement.

## Figures and Tables

**Figure 1 antibiotics-12-00606-f001:**
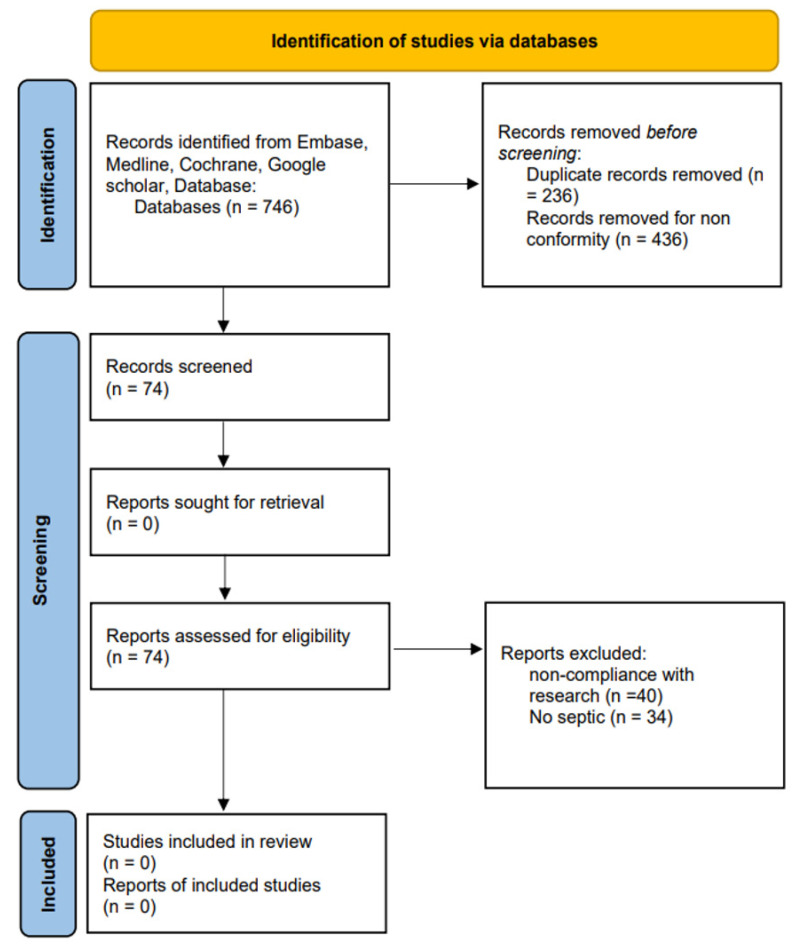
Flow chart of the screening process.

## Data Availability

No new data were created or analyzed in this study. Data sharing is not applicable to this article.

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
