# Peer review of "Antibiotic Prophylaxis in One-Stage Revision of Septic Total Knee Arthroplasty: A Scoping Review"

_antibiotics, 2023, doi:10.3390/antibiotics12030606_

Round 1
Reviewer 1 Report
Title: I would not recommend putting abbreviations in the title, better „total knee arthroplasty (TKA)“
Page 1 line 12: Periprosthetic Joint Infection should not be capitalized
Page 1 line 20: „Results: Despite our research efforts, we have not found an answer to the question which was which antibiotic to use as surgical prophylaxis for a septic revision one-stage TKA.“ You have to mention that you did not find any study to include.
page 1, lines 29 to 38: There are typographical errors throughout the manuscript. On page 1, lines 29 to 38 alone, the following stand out: „Springer et al about“ – point missing; „infection[5,8,9] .“ – space missing before brackets, space after brackets and before point; „morbidity on pa- 35
tients[3,4,5,6,7,8,9]According“ point and space missing. It goes on like this throughout the manuscript. The entire manuscript must be checked several times, preferably by different authors, for such small errors. The English language is sometimes bad.
Page 3 line 71: I am shocked that no methods are described. I understand that this is a systematic review without meta-analysis, but at least you have to explain exactly how the search was carried out. What were the inclusion and exclusion criteria? What kind of studies (design) do you look for? What were the exact mesh terms? Or rather the entire search strategy as an appendix.
Page 3 line 7: Figure 1 is nowhere mentioned in the text. Furthermore, it is damaged. I cannot read „Identification, Screening, and Included“.
Page 6 line 216: Later on page 6 the Methods section appears with some important information. How does that make any sense? How is it possible that the Results and Discussion come before the Methods?
Page 7 line 239: „all selected studies were excluded“ This is a result.
Discussion is ok, conclusions are too long.
Author Response
RESPONSE TO REVIEWER 1
Title: I would not recommend putting abbreviations in the title, better „total knee arthroplasty (TKA)“
Thank you for pointing this out. As suggested by the reviewer we explained through parentheses the meaning of the acronym TKA by adding “(total knee arthroplasty)”.
Page 1 line 12: Periprosthetic Joint Infection should not be capitalized
We agree with the reviewer’s assessment. We think this is an excellent suggestion. we proceeded to change the capitalization. turning the sentence to " One of the possible complications of this surgery is periprosthetic joint infection (PJI)”
Page 1 line 20: „Results: Despite our research efforts, we have not found an answer to the question which was which antibiotic to use as surgical prophylaxis for a septic revision one-stage TKA.“ You have to mention that you did not find any study to include.
We agree with the reviewer’s assessment. We think this is an excellent suggestion. We changed the sentence as follows:” Results: Despite our research efforts, we found no article capable of answering the question of which antibiotic to use as surgical prophylaxis for a septic revision one-stage TKA”.
page 1, lines 29 to 38: There are typographical errors throughout the manuscript. On page 1, lines 29 to 38 alone, the following stand out: „Springer et al about “– point missing; „infection[5,8,9] .“ – space missing before brackets, space after brackets and before point; „morbidity on pa- 35
tients[3,4,5,6,7,8,9]According“ point and space missing. It goes on like this throughout the manuscript. The entire manuscript must be checked several times, preferably by different authors, for such small errors. The English language is sometimes bad.
Thank you for pointing this out. As suggested by the reviewer we checked the manuscript and corrected typographical errors. We are sorry for the errors made. We have arranged for a native English speaker to read the manuscript to avoid further errors during this review.
Page 3 line 71: I am shocked that no methods are described. I understand that this is a systematic review without meta-analysis, but at least you have to explain exactly how the search was carried out. What were the inclusion and exclusion criteria? What kind of studies (design) do you look for? What were the exact mesh terms? Or rather the entire search strategy as an appendix.
Thank you for pointing this out. We appreciate the reviewer’s comment. We explained the method in the appropriate materials and methods section. In this section, we elaborated on the inclusion and exclusion methods and mesh terms used to conduct the research. In this session, we also explained the various studies we researched.
Page 3 line 7: Figure 1 is nowhere mentioned in the text. Furthermore, it is damaged. I cannot read „Identification, Screening, and Included “
Thank you for pointing this out. As suggested by the reviewer’s comment we have moved the table to the result section, going on to better explain the method by which our research was conducted. We also tried to improve the format of the table to make the words " identification, screening and included" readable.
Page 6 line 216: Later on page 6 the Methods section appears with some important information. How does that make any sense? How is it possible that the Results and Discussion come before the Methods?
Thank you for pointing this out. We are sorry for this problem. We just followed the format donated to us by the journal to provide for publication.
Page 7 line 239: „all selected studies were excluded“This is a result.
Thank you for pointing this out. As suggested by the reviewer’s comment. We have removed this sentence from the context of the study selection and integrated it into the results paragraph.
Discussion is ok, conclusions are too long.
Thank you for pointing this out. As suggested by the reviewer we proceeded to summarize part of the conclusion as follows “PJI is one of the most devastating complications of TKAs. The solution to this complication is almost always based on surgery. Based on the available data, it is not possible to make recommendations on antibiotic prophylaxis regimens for septic one stage revision TKA. Up to date, we would recommend using the same antibiotic prophylaxis as for primary joint replacement”.

Reviewer 2 Report
Thank you for submitting your article to Antibiotics. This systematic review aims to identify through a literature review, which antibiotic is used as prophylaxis for septic one-stage revision TKA and what is the rationale for its use. The authors have not found an answer to the question.
Major issues
I find that the article needs complete revision before considering potential resubmission to Antibiotics or other potential journals. A number of important points throughout the manuscript require clarification. The manuscript loses central theme too often.
1. Background:
- The aim of your study was somewhat not clear. Authors first mention prophylactic antibiotics at line 59. They only mention possible conflict between antibiotic prophylaxis and antibiotic therapy and rising antibiotic resistance. Then they immediately go to the aim of the study. The introduction before that paragraph is almost completely unnecessary. The authors should concentrate on prophylactic antibiotics, at least briefly present current guidelines and/or current questions on this topic. Not even once the current prophylactic antibiotic of choice in the primary surgery is mentioned!! Presenting the general knowledge and current problems in prophylaxis would give an introduction and the aim of the study more meaning.
- The authors should write a new introduction!
2. Methods
- Why only TKA? Why only one stage? Why not include DAIR procedure as well? Why not arthroplasty in general?
- I do not understand exclusion criteria mentioned (line 223).
- Authors asked themselves in the introduction also how prophylactic antibiotic management could act at the level of clinical outcome, however they did not define neither in the introduction nor in the methods in what kind of clinical outcome were they interested in.
- There is no necessity to include the Table A.
- Give or not to give the prophylactic antibiotic before taking the samples in one-stage or DAIR revision surgery is much more interesting topic that would bring some clinical relevance and was already discussed few times. A systematic review on this topic would be more relevant.
- Since the current aim of this systematic research does not bring any results, it would be logical, that authors, if they are interested in that topic, preform a clinical research and try to answer their question.
3. Discussion:
- Discussion is hard to read, similar problem as in introduction, there was no central theme!!
- To much of discussion is about PJI, two stage, one stage surgery, PJI diagnostics. Reading the discussion, I forgot what the main topic/question of the study was.
- Why suddenly mention empiric therapy if the question is prophylactic antibiotic (line 203)?
- New discussion needed!
In addition to major concerns, I would like to highlight a few more points:
- You should always define abbreviation (first sentence of the introduction – OA?)!
- You should always cite the literature!!!!! (line 207, line 190, line 192, line 171 …)
The manuscript cannot be published in its current form. Authors should stick to a central theme throughout the manuscript. Furthermore, the manuscript as it stands does not convince the reader that any significant conclusion can be drawn.
The authors should re-write the manuscript before resubmitting it to Antibiotics or other journal. In my opinion also a more relevant aim as suggested above is necessary.
This is certainly not intended to discourage you from the hard work which has clearly been put into this study.
Author Response
- Background:
- The aim of your study was somewhat not clear. Authors first mention prophylactic antibiotics at line 59. They only mention possible conflict between antibiotic prophylaxis and antibiotic therapy and rising antibiotic resistance. Then they immediately go to the aim of the study. The introduction before that paragraph is almost completely unnecessary. The authors should concentrate on prophylactic antibiotics, at least briefly present current guidelines and/or current questions on this topic. Not even once the current prophylactic antibiotic of choice in the primary surgery is mentioned!! Presenting the general knowledge and current problems in prophylaxis would give an introduction and the aim of the study more meaning.
We agree with the reviewer’s assessment. We think this is an excellent suggestion. We proceeded to change part of the introduction by trying to better explain the current use of prophylactic therapy based on the literature as follows from line 63 to 77. We also supplemented with other literature on the topic.
- The authors should write a new introduction!
We agree with the reviewer’s assessment. We think this is an excellent suggestion. We proceeded to change part of the introduction in line with the advice given by the reviewers as follows from line 28 to 77
- Methods
- Why only TKA? Why only one stage? Why not include DAIR procedure as well? Why not arthroplasty in general?
We agree with the reviewer’s assessment. We think this is an excellent suggestion. However, we wanted to focus the review on specific scope in order to avoid confusing factors. Certainly, the reviewer's suggestion is welcome and will be a source of discussion and stimulus for us to do more research on these other topics in the future.
- I do not understand exclusion criteria mentioned (line 223).
Thank you for pointing this out. We agree with the reviewer’s assessment. We proceeded to better explain the exclusion criteria as follow “Studies that reported clinical case reports without a validated outcome, or in which surgery was not described (to avoid bias) were excluded. Studies in which prophylaxis was not mentioned or in which prophylaxis was not described were also excluded. We also excluded all studies that involved the review of nonseptic TKA”.
- Authors asked themselves in the introduction also how prophylactic antibiotic management could act at the level of clinical outcome, however they did not define neither in the introduction nor in the methods in what kind of clinical outcome were they interested in.
Thank you for pointing this out. We agree with the reviewer’s assessment. We proceeded to change part of the introduction and rewrite the inclusion criteria to make our purpose better understood.
- There is no necessity to include the Table A.
Thank you for pointing this out. We have moved the table to the introduction section where, given the topics covered, it might be more relevant from line 40 to 45.
Give or not to give the prophylactic antibiotic before taking the samples in one-stage or DAIR revision surgery is much more interesting topic that would bring some clinical relevance and was already discussed few times. A systematic review on this topic would be more relevant.
We agree with the reviewer’s assessment. We think this is an excellent suggestion. The topic mentioned is certainly of known clinical interest. In this article we have focused on a less debated and known topic. Surely soon, accomplice also to our clinical cases treated through DAIR, we may provide a review on this other topic as well.
- Since the current aim of this systematic research does not bring any results, it would be logical, that authors, if they are interested in that topic, preform a clinical research and try to answer their question.
We agree with the reviewer’s assessment. We think this is an excellent suggestion. Our search just pointed out that there is still no scientific evidence in the literature demonstrating the superiority of one antibiotic class over another in prophylaxis for one-stage revisions of TKA. Certainly, our work will not end here. This may be the basis for starting a new multicenter clinical trial to identify the best antibiotic class for surgical prophylaxis for this type of treatment.
- Discussion is hard to read, similar problem as in introduction, there was no central theme!!
Thank you for pointing this out. We agree with the reviewer’s assessment. We proceeded to rewrite part of the discussion trying to emphasize the aim of the article from line 105 to 113.
- To much of discussion is about PJI, two stage, one stage surgery, PJI diagnostics. Reading the discussion, I forgot what the main topic/question of the study was.
Thank you for pointing this out. We agree with the reviewer’s assessment. We proceeded to change part of the discussion trying to emphasize the aim of the article from line 105 to 113. In addition, we have tried to streamline the section on surgical techniques so that we can give more space to the main topic.
- Why suddenly mention empiric therapy if the question is prophylactic antibiotic (line 203)?
Thank you for pointing this out. We agree with the reviewer’s assessment. We proceeded to change the form of the paragraph so that it was more in accordance with the construction of the full text as follows "According to le Vavasseur et al [92], the use of empiric antibiotics treatment, which usually combines a beta-lactam with activity against gram-negatives and a drug effective against gram-positives, including methicillin-resistant gram-positives, could be used at the beginning of treatment, pending culture test results and once microbiological results are available, tailor the treatment from empirical coverage to pathogen-specific treatment”. The mention of empirical therapy concerns general antibiotic therapy of PJI.
- New discussion needed!
We agree with the reviewer’s assessment. We think this is an excellent suggestion. We have changed part of the discussion by going along with the suggestions of the various reviewers. The new discussion can be read from line 93 to 171
- You should always define abbreviation (first sentence of the introduction – OA?)!
We agree with the reviewer’s assessment. We think this is an excellent suggestion. As suggested by the reviewer we explained through parentheses the meaning of the acronym OA by adding "(osteoarthritis)."
- You should always cite the literature!!!!! (line 207, line 190, line 192, line 171 …)
Thank you for pointing this out. We agree with the reviewer’s assessment. We proceeded to review the article in its entirety by going to cite the various articles that were studied.

Reviewer 3 Report
The subject of the review is quite practical and should be discussed in this issue of the journal.
Unfortunately, the authors were unable to reach a result.
However, I suggest some changes for better understanding.
1- Title: would not “capslock” and would not abbreviate TKA.
2- In line 2, I suggest clarifying the acronym OA.
3- In line 52, I suggest clarifying the acronym THA.
4- The flowchart on page 3 is untitled.
5- In line 143, I suggest clarifying the acronym IORA.
6- In line 195, I suggest clarifying the acronym DIAR.
7- I suggest that the paragraph between lines 212 to 215 could be placed at the conclusion of the article and the paragraph from lines 230 to 263 be reworded.
Author Response
1- Title: would not “capslock” and would not abbreviate TKA.
Thank you for pointing this out. As suggested by the reviewer we have changed the form of the title and we changed the form of the title by removing the “caps lock” and the abbreviation "TKA" by using " total knee arthroplasty."
2- In line 2, I suggest clarifying the acronym OA.
We agree with the reviewer’s assessment. We think this is an excellent suggestion. As suggested by the reviewer we explained through parentheses the meaning of the acronym OA by adding "(osteoarthritis)."
3- In line 52, I suggest clarifying the acronym THA
Thank you for pointing this out. As suggested by the reviewer we explained through parentheses the meaning of the acronym THA by adding “(total hip arthroplasty)”.
4-The flowchart on page 3 is untitled.
Thank you for pointing this out. We noticed that the flowchart title was lost in the layout. We have taken steps to restore it.
5- In line 143, I suggest clarifying the acronym IORA.
Thank you for pointing this out. As suggested by the reviewer we explained through parentheses the meaning of the acronym IORA by adding “(Intraosseous regional administration)”.
6- In line 195, I suggest clarifying the acronym DIAR.
Thank you for pointing this out. As suggested by the reviewer we explained through parentheses the meaning of the acronym DIAR by adding “(debridement, antibiotics, and implant retention)”.
7- I suggest that the paragraph between lines 212 to 215 could be placed at the conclusion of the article and the paragraph from lines 230 to 263 be reworded.
We agree with the reviewer’s assessment. We think this is an excellent suggestion. We placed the paragraph between the lines 212 to 215 at the conclusion of the article. We also provide to reword the paragraph from the line 230 to 263 as follows ”Potentially eligible studies were identified, and the full-text articles of these studies were retrieved for review. Reference lists of the full text articles and related citations were searched for relevant articles that were not identified in the original search. Our research strategy identified 74 studies (43 on Embase, 12 on Medline and 19 on Cochrane). PJI is one of the most devastating complications of TKAs. The solution to this complication is almost always based on surgery. Based on the available data, it is not possible to make recommendations on antibiotic prophylaxis regimens for septic one stage revision TKA. Up to date, we would recommend using the same antibiotic prophylaxis as for primary joint replacement.”

Reviewer 4 Report
General comments:
In my opinion, there are 3 important reasons why this manuscript cannot be published in its current form:
1. Quality of language is not scientific in many cases, please improve. Some examples that are inappropriate: “figure out”, “wondered about”. As a reader, language quality currently impedes publication and affects validity of the manuscript for publication in this journal.
2. It is not easy for the reader to understand the aim of this study, its main findings and how conclusions are derived. For example:
a. Introduction (lines 66-69): The purpose of this article is to find through a systematic review of the literature which antibiotics are most widely used and their rationale in the prophylaxis of one stage TKA revision due to an infection and if any, and what body of evidence support the use of specific antibiotics in one stage surgery
b. Inclusion criteria (lines 220-222): The selected population was adults with TKA infection and undergoing one-stage TKA implant revision surgery.
c. Conclusion (lines 260-262): For the time being, we would recommend using the same antibiotic prophylaxis as for primary joint replacement, i.e. cefazolin, as it was recommended for its low side effect rate and relative effectiveness. Nothing supports using only pathogen-driven antibiotics instead of wide spectrum antibiotics.
3. The validity of evidence to support any recommendation/suggestion is not sufficient and there is no analysis of outcomes. The way that authors proceeded to search and to reach their conclusions is not at all clear, and at times seems to be in contrast with standard practice (eg. what do they mean by “only pathogen-driven antibiotics”?).
Specific comments:
Title: please avoid abbreviation TKA in title.
All decimals should be the same across the manuscript (eg. 2.0% or 2.03%).
Lines 39-40: please define correctly the societies.
Lines 39-41: this small paragraph seems out of context, please consider removing or integrating better.
Lines 76-77: this statement isn’t clear. What do you mean “how many authors?”
Please provide a title for the figure and improve quality, as there is text missing inside the boxes.
I’d suggest that the search process is described in the Results, not the methods.
Author Response
RESPONSE TO REVIEWER N 4
- Quality of language is not scientific in many cases, please improve. Some examples that are inappropriate: “figure out”, “wondered about”. As a reader, language quality currently impedes publication and affects validity of the manuscript for publication in this journal.
We agree with the reviewer’s assessment. We think this is an excellent suggestion. we arranged for a native English-speaking author to proofread the manuscript to correct the various errors that were pointed out and improve the quality of the scientific language.
- It is not easy for the reader to understand the aim of this study, its main findings and how conclusions are derived. For example:
- Introduction (lines 66-69): The purpose of this article is to find through a systematic review of the literature which antibiotics are most widely used and their rationale in the prophylaxis of one stage TKA revision due to an infection and if any, and what body of evidence support the use of specific antibiotics in one stage surgery.
Thank you for pointing this out. As suggested by the reviewer we have rewritten the introduction by going further to specify the aim of the study, as follows “The purpose of this article is to find through a scoping review of the literature which class or type of antibiotic used as prophylaxis for one stage septic revision TKA might be recommend lowering the risk of reinfection, and what rationale, if any, or what body of evidence supports the use of specific antibiotics for prophylaxis of surgery in one stage revision”.
- Inclusion criteria (lines 220-222): The selected population was adults with TKA infection and undergoing one-stage TKA implant revision surgery.
Thank you for pointing this out. As suggested by the reviewer we have rewritten the inclusion criteria going to better specify our literature research for the study as follow “The study inclusion criteria consisted in studies that went on to report any form of follow-up and surveillance either face-to-face or by questionnaire or virtual methods regarding adults who had undergone one-stage revision surgery of infected TKA and in whom there was indicated antibiotic therapy”
- Conclusion (lines 260-262): For the time being, we would recommend using the same antibiotic prophylaxis as for primary joint replacement, i.e. cefazolin, as it was recommended for its low side effect rate and relative effectiveness. Nothing supports using only pathogen-driven antibiotics instead of wide spectrum antibiotics.
Thank you for pointing this out. As suggested by the reviewer we have rewritten part of the introduction from line 63 to 69 and the conclusion as follow “PJI is one of the most devastating complications of TKAs. The solution to this complication is almost always based on surgery. Based on the available data, it is not possible to make recommendations on antibiotic prophylaxis regimens for septic one stage revision TKA. Up to date, we would recommend using the same antibiotic prophylaxis as for primary joint replacement ”.
- The validity of evidence to support any recommendation/suggestion is not sufficient and there is no analysis of outcomes. The way that authors proceeded to search and to reach their conclusions is not at all clear, and at times seems to be in contrast with standard practice (eg. what do they mean by “only pathogen-driven antibiotics”?).
We agree with the reviewer’s assessment. We think this is an excellent suggestion. The final recommendation regarding the use of cefazolin is driven by the fact that while there are no specific antibiotics used for this type of surgery, the antibiotic is always essential in prophylaxis for the reasons stated above. We also proceeded to change the conclusion as above as suggested by the reviewer. The new conclusion can be read from line 200 to 204.
Title: please avoid abbreviation TKA in title.
Thank you for pointing this out. As suggested by the reviewer we have changed the abbreviation "TKA" by using " total knee arthroplasty."
All decimals should be the same across the manuscript (eg. 2.0% or 2.03%).
Thank you for pointing this out. As suggested by the reviewer we have corrected across the manuscript the shape of the percentages with the right decimals so that all are the same shape.
Lines 39-40: please define correctly the societies.
Thank you for pointing this out. As suggested by the reviewer we provided to define correctly the societies in “Musculoskeletal Infection Society (MSIS) and International Society for Infectious Diseases (ISID)”.
Lines 39-41: this small paragraph seems out of context, please consider removing or integrating better.
We agree with the reviewer’s assessment. We think this is an excellent suggestion. We proceeded to change the line to “Nowadays, many studies are still in progress to try to expand our knowledge on the diagnostics of this issue (the search for synovial biological markers is today one of the diagnostic methods with the most possible developments for future)”
Lines 76-77: this statement isn’t clear. What do you mean “how many authors?”
Thank you for pointing this out. As suggested by the reviewer we provided to change the line as follow “we found out that many authors”.
Please provide a title for the figure and improve quality, as there is text missing inside the boxes.
Thank you for pointing this out. As suggested by the reviewer we Provide to improve the quality of the figure. We also put the title of the tables in bold as per the journal's request ( line 247 and 213)
I’d suggest that the search process is described in the Results, not the methods.
Thank you for pointing this out. As suggested by the reviewer we changed the layout of the article and tried to explain the research process through Figure 1 in the results section.

Round 2
Reviewer 1 Report
Good changes.
Author Response
Thank you for your comment
Reviewer 2 Report
As authors point out, the existing literature is not amenable to a productive systematic review and that is a significant limitation of this submission.
In addition, introduction and especially discussion are hard to read and do not provide relevant information. The authors continuously talk about MSIS criteria, different surgical approaches, outcomes of the surgical approaches and the end empirical antibiotic therapy. And then suddenly they conclude which prophylactic antibiotic they would use. It is not clear how conclusions are derived. The discussion about possible different prophylactic antibiotic therapies does not exist. As a reader I get a feeling that authors forgot what their aim was and what they wanted to answer. As a reader I am not interested in how two stage or one stage revision are performed. Or how microbiological cultures are taken. Or the MSIS criteria. I do not see the point of that part of discussion at all. At least not in the current form and current aim of the study. I want to know more about prophylactic antibiotics and bacteria spectrums. Apparently, as authors stated in results section, there are no recommendations which prophylactic antibiotic should be used before septic revision TKA, however, the authors could discuss more about different prophylactic antibiotic recommendations in primary TKA and how would the authors use this knowledge and decide which prophylactic antibiotic before septic revision TKA would they use. Do authors which bacteria is the most common in TKAs (in Europe, in USA)? The resistance profile of common bacteria? For example, there is MRSA in USA and MR-CNS in Europe – both are resistant to cefazolin? Would maybe make more sense to at least use the dual antibiotic prophylaxis with vancomycin before septic revision TKA? What do studies say about that? What about the septic revisions in hip, the bacterial spectrum there seems to be different according to some studies? Any recommendation? Maybe some other dual antibiotics combinations?
Secondly, there is major grammar revision needed. I am not native speaker but following parts do not seem appropriate for a scientific article (just few examples):
OA (osteoarthritis); of IORA (Intraosseous regional administration); EOA (extended oral antibiotics); EOAP (extended oral antibiotic prophylaxis). Usually at first mention of the abbreviation you write the abbreviation in the bracket and not vice a versa.
we wondered about, line 70 – it does not seem as an appropriate phrase for a scientific article.
Line 152: …placed. von Foerster et al…. Please correct to Von (new sentence!)
Antibiotic prophylaxis vs Peri-operative antibiotic prophylaxis (PAP): chose one option and use the same one throughout the manuscript. Same applies to EOA (extended oral antibiotics) vs. EOAP (extended oral antibiotic prophylaxis)
Line 277: single-stage sepsis - sepsis? You mean revision?
Last but not least, it seems that the Table B was copied form the original article. Is that allowed?
The manuscript in current form should be completely rewritten to be even considered for a publication. The manuscript, especially discussion, in current form, is not readable and makes no sense.
I strongly recommend to the authors to read their manuscript before either re-submitting it here or to some other journal.
Author Response
RESPONSE TO REVIEWER N 3
- In addition, introduction and especially discussion are hard to read and do not provide relevant information. The authors continuously talk about MSIS criteria, different surgical approaches, outcomes of the surgical approaches and the end empirical antibiotic therapy. And then suddenly they conclude which prophylactic antibiotic they would use. It is not clear how conclusions are derived. The discussion about possible different prophylactic antibiotic therapies does not exist. As a reader I get a feeling that authors forgot what their aim was and what they wanted to answer. As a reader I am not interested in how two stage or one stage revision are performed. Or how microbiological cultures are taken. Or the MSIS criteria. I do not see the point of that part of discussion at all. At least not in the current form and current aim of the study. I want to know more about prophylactic antibiotics and bacteria spectrums. Apparently, as authors stated in results section, there are no recommendations which prophylactic antibiotic should be used before septic revision TKA, however, the authors could discuss more about different prophylactic antibiotic recommendations in primary TKA and how would the authors use this knowledge and decide which prophylactic antibiotic before septic revision TKA would they use.
Thank you for pointing this out. As suggested by the reviewer we have changed the discussion paragraph by expanding it and trying to have a more targeted focus on antibiotic prophylaxis.
- Do authors which bacteria is the most common in TKAs (in Europe, in USA)? The resistance profile of common bacteria? For example, there is MRSA in USA and MR-CNS in Europe – both are resistant to cefazolin?
We agree with the reviewer’s assessment. We think this is an excellent suggestion. As suggested by the reviewer we have expanded the discussion paragraph to cover this topic.
- Would maybe make more sense to at least use the dual antibiotic prophylaxis with vancomycin before septic revision TKA? What do studies say about that?
We agree with the reviewer’s assessment. We think this is an excellent suggestion. Unfortunately, there are no studies about single and double antibiotic prophylaxis for a septic revision of TKA. However, we expanded the discussion paragraph by also talking about double prophylaxis and its results regarding prophylaxis on primary TKA from line 183 to line 185.
- What about the septic revisions in hip, the bacterial spectrum there seems to be different according to some studies? Any recommendation? Maybe some other dual antibiotics combinations?
Thank you for pointing this out. We are grateful to the reviewer for his interest and advice. Unfortunately, the article only deals with TKA septic revision. But surely the encouragement and inspiration the reviewer has given us will be used to try to do other articles on the hip.
- OA (osteoarthritis); of IORA (Intraosseous regional administration); EOA (extended oral antibiotics); EOAP (extended oral antibiotic prophylaxis). Usually at first mention of the abbreviation you write the abbreviation in the bracket and not vice a versa.
We agree with the reviewer’s assessment. We think this is an excellent suggestion. As suggested by the reviewer we have changed the sentences, putting the abbreviation in brackets.
- we wondered about, line 70 – it does not seem as an appropriate phrase for a scientific article.
We agree with the reviewer’s assessment. We think this is an excellent suggestion. As suggested by the reviewer we have changed the sentences in” we looked at”.
- Line 152: …placed. von Foerster et al…. Please correct to Von (new sentence!)
Thank you for pointing this out. As suggested by the reviewer we have correct the sentence.
- Antibiotic prophylaxis vs Peri-operative antibiotic prophylaxis (PAP): chose one option and use the same one throughout the manuscript. Same applies to EOA (extended oral antibiotics) vs. EOAP (extended oral antibiotic prophylaxis)
Thank you for pointing this out. As suggested by the reviewer we have corrected the sentences choosing just one of the two options.
- Line 277: single-stage sepsis - sepsis? You mean revision?
Thank you for pointing this out. As suggested by the reviewer we have correct the sentence.
- Last but not least, it seems that the Table B was copied form the original article. Is that allowed?
Thank you for pointing this out. we have noticed the error. The table referring to the article below has been changed as follows line 290.

Reviewer 4 Report
The study aims and methodology are not of quality to make "recommendations". Please replace "recommend" with "suggest".
Despite authors' efforts to improve the manuscript, the scientific evidence provided by this manuscript remains low.
The quality of English remains of very low level. Authors responded that an English proficient person will review the manuscript. Whether the manuscript can be accepted in this form and quality, is up to the Editor.
Author Response
RESPONSE TO REVIEWER N 4.2
- The study aims and methodology are not of quality to make "recommendations". Please replace "recommend" with "suggest".
Thank you for pointing this out. As suggested by the reviewer we changed the phrase in “Up to date, we would suggest using the same antibiotic prophylaxis as for primary joint replacement.”
- Despite authors' efforts to improve the manuscript, the scientific evidence provided by this manuscript remains low.
Thank you for pointing this out. As suggested by the reviewer we have tried to improve the quality of the discussion and increase the quality and scientific evidence of this article.
- The quality of English remains of very low level. Authors responded that an English proficient person will review the manuscript. Whether the manuscript can be accepted in this form and quality, is up to the Editor.
We agree with the reviewer’s assessment. We think this is an excellent suggestion. we arranged for a native English-speaking author to proofread the manuscript to correct the various errors that were pointed out and improve the quality of the scientific language.

Round 3
Reviewer 4 Report
I'd like to thank the authors who made significant effort to respond to my comments.